# OpenReview forum: "Learning Diverse Attacks on Large Language Models for Robust Red-Teaming and Safety Tuning"
_ICLR.cc/2025/Conference — ICLR 2025 Poster_

### Official Review · Reviewer_YDLU · 2024-10-28

**Soundness:** 3
**Presentation:** 4
**Contribution:** 4
**Rating:** 8
**Confidence:** 2

**Summary:**

This paper introduces GFlowNet fine-tuning plus a follow-up smoothing phase for attack generation in LLM red teaming. Their approach overcomes the typical problems of lacking diversity, effectiveness and model collapse that arise in RL-based automated red teaming. The authors not only show the effectiveness of their method for red teaming (toxicity focus) but also that their method can be used to generate highly effective fine-tuning data for safety tuning.

**Strengths:**

The method is well motivated, presented in an easy to understand way, and backed by a rigorous experimental setup. Especially the performance in the transfer setting is impressive, as most other methods completely fail at this task. This paper advances the state of the art in a significant way and addresses a crucial problem in AI security.

**Weaknesses:**

Experiments on the performance against LLM guardrails would have been of interest, as real-life deployments will always include input and output guardrail models. Given the strong performance of the method in transfer settings, this could also prove to be another potential strongpoint of this method.

**Questions:**

Could you please elaborate on how you expect your method to fare against current SOTA guardrails and the challenges you see in overcoming those with attacks using your method?

---

> ### Author Response · Authors · 2024-11-15
> **Pointers to specific guardrails**
>
> Dear Reviewer YDLU,
>
> We thank you for your thoughtful comments and feedback. As we prepare our rebuttal we would be grateful if you could point us to any particular state-of-the-art guardrails you have in mind. At the same time, we would like to note that typically red-teaming is done _before_ release to ensure the harmful behaviours can be identified before deployment within a larger system with guardrails.
>
> Thanks,
>
> Authors

---

> > ### Comment · Reviewer_YDLU · 2024-11-17
> >
> > Dear authors, I understand that red-teaming is typically done before release (and thus I also gave you a high rating for the paper). Incorporating guardrails into the pre-deployment evaluation (e.g. Llama Guard) would be an interesting step to increase the real-life impact of the work. I do not expect you to add this to your rebuttal, as I see the paper as an accept the way it is. Good luck.

---

> ### Author Response · Authors · 2024-11-21
> **Author response to Reviewer YDLU**
>
> Thank you for your positive assessment and quick response to our question!
>
> We agree that incorporating guardrails in the pre-deployment red-teaming would be an interesting scenario for our approach. Let us consider a setup where the target language model (LM) is coupled with a Llama-Guard model which evaluates the responses generated by the LM and discards responses which cross some threshold of toxicity. As our approach only assumes black-box access to the system, in principle we can target this complete system directly. This would require a different reward model to score the responses as harmful. Additionally, combined with the guardrail, we expect it to be harder to elicit harmful responses by the LM with the guardrails -- making the training of the GFlowNet policy in stage 1 more challenging. We believe this could be an interesting avenue to explore in future work.

---

> > ### Comment · Reviewer_YDLU · 2024-11-25
> >
> > Thank you for the additional info on future work. I am looking forward to the follow-up paper to this work.

---

### Official Review · Reviewer_pUvQ · 2024-10-31

**Soundness:** 3
**Presentation:** 3
**Contribution:** 2
**Rating:** 8
**Confidence:** 4

**Summary:**

The paper studies methods of training a language model to generate adversarial prompts that, when provided to more standard language models (such as GPT, Gemma, or Llama), produce responses deemed to be violating by a safety classifier. The main contribution of the paper is to apply the GFlowNet reinforcement learning method for this fine tuning of the attacker model. The paper produces attacker models that generate prompts of better diversity and higher attack success rate than other methods that take the same approach to red teaming.

**Strengths:**

On a standardized dataset and with standardized evaluations, and within the class of reinforcement learning methods that train an attacker model to produce a single adversarial prompt, this paper proposes a method that does achieve better diversity and stronger attack success rate. It is important for the community to be aware that this reinforcement learning method can produce attacker models with stronger adversarial “power” and diversity. The discussion of the adjustments needed to the GFlowNet method is also important.

**Weaknesses:**

I would say that this paper suffers from weaknesses that the whole field of automated jailbreaking is currently subject to. I do not consider this to be a reason for rejection but I also think it is important to record this critique for future iterations of automated red teaming work.
To be more precise, it is not clear to me that the majority of methods in this field – this work included – discover ways to elicit meaningfully harmful responses from models. It appears to me that the responses provided in the Appendix are all generally informational. To put this another way, this paper – along with most works who take AdvBench off the shelf as the goal set and slap some classifier on the responses – sidestep a robust definition of what is and is not considered harmful. This is not necessarily a problem of the work itself and rather a shortcoming of the yardsticks (harmful prompt datasets and classifiers) that have been widely adopted.

However, what is important for the authors of this work, is that the end result of this leads to methods that generate adversarial prompts that likely exploit the model’s willingness to be generically helpful. In particular, the prompts listed in tables B.5 and B.6 are themselves very generic. For example, it is not clear why or how “records in the financial database” are being manipulated. Is this necessarily some harmful or unauthorized manipulation? The model response itself assumes that you would be doing this with a properly authorized user. This is likely because the prompt leaves enough vagueness in it to be interpreted as asking for something that is perfectly legal and acceptable (help with interacting with a database).

Thus, I believe methods in this space in the future should also consider specificity of the harmful request as another axis of desirable properties, in addition to diversity and attack success rate judged by a classifier. So instead of ending up with prompts that dance around a vaguely specified line, methods should a) be explicit about the line and b) make sure that their attacks clearly cross it. It would be interesting if GFlowNet adversarial methods can help elicit specific and truly harmful responses from language models.

**Questions:**

On linea 305-309, it is not clear to me what dataset is being used. Can you be more explicit if you are adopting the datasets for all methods listed on these lines? Or do you use a different dataset for your fine tuning?

---

> ### Author Response · Authors · 2024-11-21
> **Author response to Reviewer pUvQ**
>
> Thank you for you review and positive assessment of the paper.
>
> We appreciate your thorough critique on the general setup of automated red-teaming. We agree with your suggestion about incorporating the specificity of the attack as a desiderata. In principle, our approach can incorporate such specificity given access to a classifier which can classify attacks and the target response's adherence to a given harm specification. Unfortunately, from a preliminary investigation with Llama-Guard, it appears that toxicity classifiers struggle with reliable classification of attacks and generated responses.
>
> As we note in the conclusion ("Limitations" paragraph), our proposed method is only as good as the toxicity classifier used, and is thus limited by the failures of the knowledge represented in it. On the other hand, diversity-seeking methods such as the one we propose are provably less sensitive to reward misspecification [1,2] and could be less vulnerable to spurious modes of the classifier.
>
> > On line 305-309, it is not clear to me what dataset is being used. Can you be more explicit if you are adopting the datasets for all methods listed on these lines? Or do you use a different dataset for your fine tuning?
>
> For PPO, REINFORCE and GFlowNet, the initial policy is the SFT model (trained on SafetyDataset and AdvBench). For GFlowNet + MLE, the policy is also initialized with the SFT model, but trained using the samples generated during the training of the GFlowNet.
>
> ### References
>
> [1] Eysenbach, Benjamin, et al. "Maximum entropy RL (provably) solves some robust RL problems", ICLR 2022.
>
> [2] Gao, Leo, et al. "Scaling laws for reward model overoptimization", ICML 2023.
>
> **Thank you again for the interesting comments. Please let us know if we can provide further clarifications.**

---

> > ### Comment · Reviewer_pUvQ · 2024-11-25
> >
> > Thank you for the response and the clarification! I will keep my original score and recommend an accept.

---

### Official Review · Reviewer_3L4U · 2024-11-03

**Soundness:** 3
**Presentation:** 3
**Contribution:** 2
**Rating:** 6
**Confidence:** 4

**Summary:**

The paper proposes a two-stage approach combining GFlowNet fine-tuning with MLE smoothing to automatically generate diverse and effective adversarial prompts for testing language model safety, demonstrating good performance over a selection of baseline red teaming methods, balancing attack success rate and prompt diversity while enabling reasonable transfer to new target models.

**Strengths:**

1. The proposed method could sample diverse prompts and appear more efficient than the baseline methods.
2. The main claims are well-supported by the experiments.
3. The writing is clear and the paper is easy to follow.

**Weaknesses:**

1. There's a lack of evaluation against stronger attacks and defenses. The paper did not consider many of the non-optimization-based gray /black box attacks, which might perform better at lower computation budget. The paper also did not consider robustified versions of the models such as models that underwent circuit breaking or (latent) adversarial training.
2. It's unclear how to adapt this for targeted attack against specific harmful requests and how well that would work.

**Questions:**

I'm curious to know whether the method scales with stronger base model for the attacker model.

**Details Of Ethics Concerns:**

No ethics concerns

---

> ### Author Response · Authors · 2024-11-21
> **Author response to Reviewer 3L4U**
>
> Thank you for your helpful comments and questions. We have done our best to answer them below.
>
> > lack of evaluation against stronger attacks and defenses. The paper did not consider many of the non-optimization-based gray /black box attacks, which might perform better at lower computation budget.
>
> We are not sure what non-optimization approaches you are referring to. We would greatly appreciate it if you could point us to any baseline in particular you would like us to compare to.
>
> > The paper also did not consider robustified versions of the models such as models that underwent circuit breaking or (latent) adversarial training.
>
> Thank you for the comment. We applied our approach to [Llama-2-7B-CAT](https://huggingface.co/ContinuousAT/Llama-2-7B-CAT) and [Llama-3-8B-Instruct-RR](https://huggingface.co/GraySwanAI/Llama-3-8B-Instruct-RR). The Llama-2-7B-CAT model is Llama-7b based model, which is adversarially tuned to defend against jailbreak attacks [1]. Llama-3-8B-Instruct-RR is trained with circuit breaking for robustness against jailbreak attacks. We achieve attack success rates of 80.07% and 65.55%, for Llama-2-7B-CAT and Llama-3-8B-Instruct RR, respectively, based on the reward model, Llama-Guard-3. However, upon closer inspection we found that the Llama-Guard 3 reward model was giving high scores to refusals as well. This is one of the challenges we discussed in the paper: our method is only as good as the reward model, which fails in this case. We believe there could be interesting future work to alleviate these challenges
>
>
> > It's unclear how to adapt this for targeted attack against specific harmful requests and how well that would work.
>
> In principle, the GFlowNet fine-tuning based method is applicable for jailbreaking-style attacks. To see how this works, we can view jailbreaking as an infilling problem where the harmful query is beginning of the text and the positive response from the language model is the end, and the prompt suffix is to be infilled. [3] demonstrated GFlowNet fine-tuning for this setting. We believe this is an interesting avenue for future work.
>
> > I'm curious to know whether the method scales with stronger base model for the attacker model.
>
> Thanks for the comment. We expect the method to produce stronger attacks as we scale the base model. We did a small experiment using Llama3.2-Instruct-1B as the base model for the attacker targeting Llama-3.1-Instruct-8B. As shown in Table R.2 below, the attack with Llama-3.2-Instruct-1B significantly improves the ratio of toxic prompts, while retaining a similar level of diversity.
>
> **Table R.2**: The attacker model with Llama-3.2-1B-Instruct to target Llama-3.1-8B-Instruct.
> | Base Model   | Ratio of Toxic Prompts (%) |  Cosine Distance |
> |--------------|:--------------------------:|:----------------:|
> | GPT2-Small   |       $81.05\pm 0.96$      | $0.829\pm 0.001$ |
> | Llama-3.2-1B |       $\textbf{92.71}\pm 1.12$      |  $0.833\pm0.002$ |
>
> ### References
> [1] Xhonneux, Sophie, et al. "Efficient adversarial training in llms with continuous attacks." NeurIPS 2024.
>
>
> [2] Zou, Andy, et al. "Improving alignment and robustness with circuit breakers." NeurIPS 2024.
>
> [3] Hu, Edward J., et al. "Amortizing intractable inference in large language models", ICLR 2024.
>
> **Thank you again for review, and do not hesitate to let us know if we can provide further clarifications.**

---

> > ### Author Response · Authors · 2024-11-25
> >
> > Dear Reviewer 3L4U,
> >
> > We thank you again for your helpful feedback and hope that our rebuttal addressed your concerns and comments. We believe we have answered all of your original questions with our response and new experiment with a stronger base model above.  As the end of the discussion period draws closer, we would like to ask if you have any further questions or comments on our work. We would be happy to address them.

---

> ### Author Response · Authors · 2024-12-02
> **Gentle Reminder**
>
> Dear Reviewer 3L4U,
>
> This is just a gentle reminder to consider our responses above and the new experiments we conducted. If these lead you to favour acceptance of our paper, we would be grateful if you could update your review accordingly.
>
> Thanks,
>
> The authors.

---

> > ### Comment · Reviewer_3L4U · 2024-12-02
> >
> > Thank you for the additional experiments. I have raised the score by 1, though some of my concerns are still unresolved.

---

### Official Review · Reviewer_UQkL · 2024-11-04

**Soundness:** 3
**Presentation:** 3
**Contribution:** 2
**Rating:** 5
**Confidence:** 3

**Summary:**

The paper proposes an automatic two step red-teaming method for LLMs that aims at generating prompts that are very diverse as well as effective. Diversity is important to transfer attacks across different models and for generalization purposes. The first step of the red-teaming process consists of collecting several prompts that score a high reward using GFlowNets algorithms. In these methods, the learned policy sample prompts with probability proportional to the reward the prompts received. The reward is defined as the likelihood that a prompt is classified as toxic by a toxicity classifier (plus a KL term). The second step takes into account some issues related to tuning some hyperparameters in the reward function. In particular, the prompts collected at the previous step are filtered and used to fine-tune the initial policy to maximize model log-likelihood. The red-teaming is tested against different baselines and across different models. The results show that prompts generated with this method are more diverse and effective compared to the ones produced by other baselines and transfer across different models.

**Strengths:**

- Attack diversity is important in red-teaming and finding automatic methods to produce diverse and effective prompts is valuable.
- The second step seems to significantly improve over the baseline consisting only of the first step.
- The method is compared against many different baselines.

**Weaknesses:**

- The tested models are quite small. Even if the attacks generalize to different models, it’d be good to have an evaluation of bigger and closed-source models (since the technique doesn’t require access to model weights).
- Although inexpensive, it seems not very efficient to retrain the policy on a subset of generated prompts. For example, filtering the prompt can be done on the fly during the GFlowNet stage.

**Questions:**

- Why do you use two different toxicity classifiers for different models?
- The difference in performance between GFlowNets and GFlowNets+MLE changes quite a bit across different models. For example, for Dolly and GPT-2 the performance is almost the same, for Gemma there is a huge gap, for the Llamas the difference is a bit smaller. Do you have a sense of why this is happening?
- The attacks produced with this method are very diverse. Is this diversity mainly semantic or also stylistic?

---

> ### Author Response · Authors · 2024-11-15
> **Review for the wrong paper?**
>
> Dear Reviewer UQkL,
>
> We believe there may have been a mistake, since the review you posted here does not appear to be about our paper. We ask you to please check if this may been an error.
>
> Thanks,
>
> Authors

---

> > ### Comment · Reviewer_UQkL · 2024-11-16
> > **Re: Review for the wrong paper?**
> >
> > Hi, thank you for flagging this. I accidentally submitted my review for another paper to your submission, and my review for your paper to their submission. I've contacted the area chairs to get the reviews swapped to their correct submissions. My apologies for any confusion this caused.

---

> ### Author Response · Authors · 2024-11-16
> **Re: Re: Review for the wrong paper?**
>
> Dear Reviewer UQkL,
>
> Thank you for your prompt response! We would like to note that the reviews can still be edited at this time. Could you kindly update your review here with any feedback on our paper? This would allow us some time to address any concerns you might have.
>
> Thank you,
>
> Authors

---

> > ### Comment · Reviewer_UQkL · 2024-11-17
> > **Re: Review for the wrong paper?**
> >
> > I've now updated my review of your paper. Thank you again for letting me know about the error.

---

> ### Author Response · Authors · 2024-11-21
> **Author response to Reviewer UQkL**
>
> Thank you for your comments. We appreciate that you quickly updated the content to give us time to respond. Below we have done our best to answer the points you made, including a new experiment showing **good transfer performance to attack GPT-4o**.
>
> > The tested models are quite small. Even if the attacks generalize to different models, it’d be good to have an evaluation of bigger and closed-source models (since the technique doesn’t require access to model weights).
>
> Since our method involves on-policy training in the first stage, running an experiment with a closed model as the target would be expensive and not feasible for us. However, we would like to note that in our transfer experiments (summarized in Table 2), we do consider larger models, such as Llama-2-70B-chat and Llama-3-70B-instruct. Additionally, we generate 1,024 prompts originally targeted at Llama-2-7B-chat and transfer them to attack GPT-4o. Across five random seeds, **an average of 65% of the prompts elicit harmful responses from GPT-4o**.
>
>
> > Although inexpensive, it seems not very efficient to retrain the policy on a subset of generated prompts. For example, filtering the prompt can be done on the fly during the GFlowNet stage.
>
> As we show in Table 3, the second stage accounts for a very small fraction of total training time. While it could be possible to train the second-stage SFT model "online", this would present two challenges. First, it would require keeping two copies of an attacker model in memory (the GFlowNet policy and the second-stage SFT attacker policy). Second, the samples used for SFT would not be seen in random order, but in the order they were discovered by the GFlowNet sampler, possibly leading to bias or catastrophic forgetting.
>
> > Why do you use two different toxicity classifiers for different models?
>
> For preliminary experiments, following the work of [1], we use the RoBERTa-based toxicity classifiers for GPT-2 and Dolly. However, when red-teaming target large language models with safety alignment, the classifier assigns unreasonably high toxicity score to refusal responses to attack prompts, as shown in Table B.1. As a result, we switch it to Llama-Guard for red-teaming Llama and Gemma.
>
> > The difference in performance between GFlowNets and GFlowNets+MLE changes quite a bit across different models. For example, for Dolly and GPT-2 the performance is almost the same, for Gemma there is a huge gap, for the Llamas the difference is a bit smaller. Do you have a sense of why this is happening?
>
> The intensity of safety alignment in target language models makes a significant difference. GPT-2 and Dolly lack safety guardrails and rarely refuse to respond to prompts, making the discovery of attack prompts relatively easy. On the other hand, the target language models with strong safety alignment, such as Llama-2-7b-chat and Gemma-2b-it, often refuse to answer to naive attack prompts, resulting in a sparse reward landscape. As discussed in lines 53-65, this sparsity poses a significant challenge for GFlowNet, making it extremely difficult to balance the trade-off between diversity and toxicity.
>
> > The attacks produced with this method are very diverse. Is this diversity mainly semantic or also stylistic?
>
> Based on our qualitative study on a small subset of generated prompts, the diversity is primarily semantic.
>
> ### References
>
> [1] Hong, Zhang-Wei, et al. "Curiosity-driven Red-teaming for Large Language Models", ICLR 2024.
>
> **Thank you again for your comments, questions, and suggestions, and do not hesitate to tell us if we can provide further answers.**

---

> > ### Author Response · Authors · 2024-11-25
> >
> > Dear Reviewer UQkL,
> >
> > We thank you again for your helpful feedback and hope that our rebuttal addressed your concerns and comments. We believe we have answered all of your original questions, and we’ve additionally tested the transfer performance to a much larger and stronger target model (GPT-4o).  As the end of the discussion period draws closer, we would like to ask if you have any further questions or comments on our work. We would be happy to address them.

---

> ### Author Response · Authors · 2024-12-02
> **Gentle Reminder**
>
> Dear Reviewer UQkL,
>
> This is just a gentle reminder to consider our responses above and the new experiments we conducted. If these lead you to favor acceptance of our paper, we would be grateful if you could update your review accordingly.
>
>
> The authors.

---

> > ### Comment · Reviewer_UQkL · 2024-12-02
> >
> > Thank you for your responses and the additional GPT-4o experiments. Your rebuttal has addressed my questions. I have no further concerns and will maintain my current rating.

---

### Official Review · Reviewer_nedW · 2024-11-05

**Soundness:** 4
**Presentation:** 4
**Contribution:** 3
**Rating:** 8
**Confidence:** 4

**Summary:**

This paper applies GFlowNet to the problem of automated red-teaming, achieving a favorable balance between attack success rate and attack diversity. A model is trained to sample attacks with probability proportionally to their reward, followed by a cheaper second stage of training a smoother version of the model on rollouts of the trained model. The method achieves high levels of diversity combined with strong attack success rates, and, when including the smoothing step, consistently strong results across different hyper-parameters and different target models.

**Strengths:**

- Clearly strong results and exciting possibilities for improving automated red-teaming and finding diverse attacks with strong motivations as one of the core challenges.
- The paper is well written and easy to follow. Experiments and ablations are documented well and replication of the results seems straightforward.
- I appreciate comparing to multiple RL-based baselines, as well as red-teaming multiple models. This gives confidence that the results will hold up in a wider range of settings.

**Weaknesses:**

- Given the focus on learning diverse attacks, there could have been more in-depth ablations and experiments to investigate which parts of the method most strongly influence diversity (also see Questions section).
- It would be nice to also plot error bars, at least on a version in the appendix. It was not clear to me if the various plots are also based on multiple random seeds (as table 2 is).

**Questions:**

- Where does the diversity come from (besides the ablation on $\beta$ values)? $\gamma$? sampling temperature $\tau$? replay buffer / "off-policy-ness"? The entropy/diversity of $p_{ref}$? The mix between replay buffer and online sampling in each iteration? Given that this is a main focus of the paper I would have been excited about more ablations / investigations in this direction.
- The results for REINFORCE, ICL, SFT seem mostly consistent across the different red-teamed models. PPO+Novelty results are less consistent and so is GFlowNet - is this due to hyper-parameter sensitivity? Or variance between runs? GFlowNet+MLE looks more consistently strong, so the core results of the paper are not impacted by this.
- How strong was the toxicity classifier, how often did the method "hack" shortcomings of the classifier rather than finding actual toxic outputs? This was hinted at in the discussion of setting $\beta$ too low (high weight of R1), but wondering if there are some more concrete results on this?
- What are sensible values for $r1, r2$?

---

> ### Author Response · Authors · 2024-11-21
> **Author response to Reviewer nedW (1/2)**
>
> Thank you for your review, the positive assessment of the paper, and the interesting questions. We've answered the points you raised below.
>
> > Where does the diversity come from (besides the ablation on $\beta$ values)? $\gamma$? sampling temperature $\tau$? replay buffer / "off-policy-ness"? The entropy/diversity of $p_\texttt{ref}$? The mix between replay buffer and online sampling in each iteration? Given that this is a main focus of the paper I would have been excited about more ablations / investigations in this direction.
>
> This is a great point. The key source of diversity in our approach is the GFlowNet fine-tuning objective which trains the attacker to sample from the reward distribution. Prior work [1,2] has demonstrated that sampling from the reward distribution is effective for generating diverse and high reward samples. As you point out there are several design choices which can be ablated, but we focus on the $\beta$ as that is particularly relevant in our setting and due to limited compute resources. Several of the other parameters you mention have been studied in prior work (e.g. [3,4,5]).
>
> > It would be nice to also plot error bars, at least on a version in the appendix. It was not clear to me if the various plots are also based on multiple random seeds (as table 2 is).
>
> Thank you for the suggestion. In Appendix B.5, following your suggestion, we have included standard deviation as well as average of five different experimental runs of the results for the transfer exepriments and safety fine-tuning.
>
> > The results for REINFORCE, ICL, SFT seem mostly consistent across the different red-teamed models. PPO+Novelty results are less consistent and so is GFlowNet - is this due to hyper-parameter sensitivity? Or variance between runs? GFlowNet+MLE looks more consistently strong, so the core results of the paper are not impacted by this.
>
> In our experiments, PPO + Novelty and GFlowNet both struggle to attack target language models with strong safety guardrails, such as Llama-2-7b-chat and Gemma-2b-it. These strong guardrails make the reward sparse, leading to difficulty in tuning the hyperparameters that control trade-off between diversity and toxicity (reward temperature $\beta$ and $\gamma$ for GFlowNet or the weight of the novelty reward for Novelty + PPO). In contrast, our GFlowNet + MLE shows less sensitivity to the hyperparameter choices, as shown in Figure 5.
>
> > How strong was the toxicity classifier, how often did the method "hack" shortcomings of the classifier rather than finding actual toxic outputs? This was hinted at in the discussion of setting too low (high weight of R1), but wondering if there are some more concrete results on this?
>
>
> In the preliminary experiments, following the work of [6], we use the RoBERTa-based toxicity classifier [7]. However, as shown in Table B.1, the model assigns high toxicity score to the response that refuses to answer the prompt. After switching it to Llama-Guard, we have manually checked some of prompts and found no longer observe such false positives in our main experiments.
> As shown in Table B.7, Table B.8, and Table B.9, Llama-Guard assigns low toxicity score to the refusal responses, such as "I can’t help with that. Is there anything else I can help you with?".
>
> When conducting additional experiments suggested by Reviewer 3L4U,  we observed even Llama-Guard assigned high toxicity score to refusal responses from [Llama-2-7B-CAT](https://huggingface.co/ContinuousAT/Llama-2-7B-CAT) [8] and [Llama-3-8B-Instruct-RR](GraySwanAI/Llama-3-8B-Instruct-RR) [9], which are Llama model families  adversarially trained to defend against jailbreak attacks. This is one of the challenges we discussed in the paper: our method, like other RL-based approaches, is only as good as the reward model, which fails in this case. We believe there could be interesting future work to alleviate these challenges.

---

> ### Author Response · Authors · 2024-11-21
> **Author response to Reviewer nedW (2/2)**
>
> > What are sensible values for $r_1, r_2$?
>
> As specified in Appendix A, we set $\exp(r_1)$ to $0.7$ and $r_2$ to $-100$.
> Moreover, varying the toxicity threshold, we filter the prompts discovered at the first stage and fine-tune the attacker model on the chosen prompts to red-team the Llama-2-7b model. As shown in the table below, toxicity rate tends to increase, while diversity tends to decrease, as the filter becomes more selective for high-reward prompts. We expect that with increasing threshold, the size of the training data for Stage 2 will decrease, leading to overfitting and worse transfer performance.
>
> **R.1** Experiments with different toxicity threshold.
> | Toxicity Threshold $(e^{r_1})$   | Toxicity Rate (%) | Cosine Distance |
> |:---------------------|:-----------------:|:---------------:|
> | 0.0 (SFT)            | $\phantom{0}0.00$ |       $0.86$    |
> | 0.1                  |       $28.71$     |       $0.72$    |
> | 0.2                  |       $36.13$     |       $0.73$    |
> | 0.3                  |       $40.91$     |     $0.75$      |
> | 0.4                  |       $45.60$     |     $0.72$      |
> | 0.5                  |       $51.36$     |     $0.70$      |
> | 0.6                  |       $56.44$     |     $0.70$      |
> | 0.7                  |       $62.71$     |     $0.69$      |
> | 0.8                  |       $77.92$     |     $0.66$      |
> | 0.9                  |       $87.98$     |     $0.63$      |
>
>
> ### References
>
> [1] Bengio, Emmanuel, et al. "Flow network based generative models for non-iterative diverse candidate generation." NeurIPS 2021.
>
> [2] Hu, Edward J., et al. "Amortizing intractable inference in large language models", ICLR 2024.
>
> [3] Atanackovic, Lazar, and Emmanuel Bengio. "Investigating Generalization Behaviours of Generative Flow Networks." arXiv preprint arXiv:2402.05309.
>
> [4] Vemgal, Nikhil, Elaine Lau, and Doina Precup. "An empirical study of the effectiveness of using a replay buffer on mode discovery in gflownets." arXiv preprint arXiv:2307.07674.
>
> [5] Shen, Max W., et al. "Towards understanding and improving gflownet training." ICML 2023.
>
> [6] Hong, Zhang-Wei, et al. "Curiosity-driven Red-teaming for Large Language Models", ICLR 2024.
>
>
> [7] Vidgen, Bertie, et al. "Learning from the Worst: Dynamically Generated Datasets to Improve Online Hate Detection", ACL 2021.
>
> [8] Xhonneux, Sophie, et al. "Efficient adversarial training in llms with continuous attacks." NeurIPS 2024.
>
> [9] Zou, Andy, et al. "Improving alignment and robustness with circuit breakers." NeurIPS 2024.
>
> **Thank you again for your comments. We are happy to answer any further questions you have during the discussion period.**

---

### Author Response · Authors · 2024-11-21
**General response to all reviewers**

We express our sincere gratitude to all the reviewers for their constructive comments and feedback and generally positive reception of our work.

We particularly appreciate their recognition of the **strong experimental results** (nedW, 3L4U, pUvQ, UQkL), **strong motivations** (nedW, YDLU), and **high quality of writing** (nedW, 3L4U, YDLU).


**Additional experimental results**
We would like to highlight some additional experiments we performed based on suggestions by the reviewers.

- **Transfer to GPT-4o**: Based on a suggestion by Reviewer UQkL, we transfer the attack prompts, which originally targeted the Llama-2-7b-chat model, to GPT-4o and evaluate how many of them can elicit harmful responses from GPT-4o. In average of 5 different runs, **65% of 1,024 prompts can successfully attack GPT-4o**.

- **Attackers with larger models**: Suggested by Reviewer 3L4U, we train the attacker model using Llama-3.2-1B-Instruct to red-team the Llama-3.1-8B-instrcut model. As shown in the table below, the larger model achieves a larger number of toxic prompts than smaller model GPT2-Small.


| Base Model   | Success rate (%) |  Cosine Distance |
|--------------|:--------------------------:|:----------------:|
| GPT2-Small   |       $81.05\pm 0.96$      | $0.829\pm 0.001$ |
| Llama-3.2-1B |       $\textbf{92.71}\pm 1.12$      |  $0.833\pm0.002$ |


We have responded to all the individual comments from the reviewers below. Please let us know if you have any further questions or suggestions!

---

### Meta-Review · Area_Chair_B573 · 2024-12-24

**Metareview:**

This paper applies GFlowNet to the problem of automated red-teaming, reaching a favorable balance between attack success rate and attack diversity. The authors are encouraged to include scalable evaluation and thorough results analysis in the final version.

**Additional Comments On Reviewer Discussion:**

The reviewers agree with the final decision.

---

### Decision · Program_Chairs · 2025-01-22

Accept (Poster)